# Creation of a 3D Goethite–Spongin Composite Using an Extreme Biomimetics Approach

**DOI:** 10.3390/biomimetics8070533

**Published:** 2023-11-09

**Authors:** Anita Kubiak, Alona Voronkina, Martyna Pajewska-Szmyt, Martyna Kotula, Bartosz Leśniewski, Alexander Ereskovsky, Korbinian Heimler, Anika Rogoll, Carla Vogt, Parvaneh Rahimi, Sedigheh Falahi, Roberta Galli, Enrico Langer, Maik Förste, Alexandros Charitos, Yvonne Joseph, Hermann Ehrlich, Teofil Jesionowski

**Affiliations:** 1Faculty of Chemistry, Adam Mickiewicz University, Uniwersytetu Poznanskiego 8, 61-614 Poznan, Poland; markot6@amu.edu.pl (M.K.); barles5@amu.edu.pl (B.L.); 2Center of Advanced Technology, Adam Mickiewicz University, Uniwersytetu Poznanskiego 10, 61-614 Poznan, Poland; mpszmyt@amu.edu.pl (M.P.-S.); herehr@amu.edu.pl (H.E.); 3Institute of Electronics and Sensor Materials, TU Bergakademie Freiberg, Gustav-Zeuner-Str. 3, 09599 Freiberg, Germany; voronkina@vnmu.edu.ua (A.V.); parvaneh.rahimi@esm.tu-freiberg.de (P.R.); sedigheh.falahi@doktorand.tu-freiberg.de (S.F.); yvonne.joseph@esm.tu-freiberg.de (Y.J.); 4Department of Pharmacy, National Pirogov Memorial Medical University, Vinnytsya, Pyrogov Street 56, 21018 Vinnytsia, Ukraine; 5IMBE, CNRS, IRD, Aix Marseille University, Station Marine d’Endoume, Rue de la Batterie des Lions, 13007 Marseille, France; alexander.ereskovsky@imbe.fr; 6Institute of Analytical Chemistry, TU Bergakademie Freiberg, Leipziger Str. 29, 09599 Freiberg, Germany; korbinian.heimler@chemie.tu-freiberg.de (K.H.); anika.rogoll@chemie.tu-freiberg.de (A.R.); carla.vogt@chemie.tu-freiberg.de (C.V.); 7Department of Medical Physics and Biomedical Engineering, Faculty of Medicine Carl Gustav Carus, TU Dresden, Fetscherstr. 74, 01307 Dresden, Germany; roberta.galli@tu-dresden.de; 8Institute of Semiconductors and Microsystems, TU Dresden, Nöthnitzer Str. 64, 01187 Dresden, Germany; 9Institute for Nonferrous Metallurgy and Purest Materials (INEMET), TU Bergakademie Freiberg, Leipziger Str. 34, 09599 Freiberg, Germany; maik.foerste@inemet.tu-freiberg.de (M.F.); alexandros.charitos@inemet.tu-freiberg.de (A.C.); 10Faculty of Chemical Technology, Institute of Chemical Technology and Engineering, Poznan University of Technology, Berdychowo 4, 60-965 Poznan, Poland

**Keywords:** extreme biomimetics, spongin, goethite, 3D scaffold, composite, sensor, sponges

## Abstract

The structural biopolymer spongin in the form of a 3D scaffold resembles in shape and size numerous species of industrially useful marine keratosan demosponges. Due to the large-scale aquaculture of these sponges worldwide, it represents a unique renewable source of biological material, which has already been successfully applied in biomedicine and bioinspired materials science. In the present study, spongin from the demosponge *Hippospongia communis* was used as a microporous template for the development of a new 3D composite containing goethite [α-FeO(OH)]. For this purpose, an extreme biomimetic technique using iron powder, crystalline iodine, and fibrous spongin was applied under laboratory conditions for the first time. The product was characterized using SEM and digital light microscopy, infrared and Raman spectroscopy, XRD, thermogravimetry (TG/DTG), and confocal micro X-ray fluorescence spectroscopy (CMXRF). A potential application of the obtained goethite–spongin composite in the electrochemical sensing of dopamine (DA) in human urine samples was investigated, with satisfactory recoveries (96% to 116%) being obtained.

## 1. Introduction

Biomimetics is a field of modern research that seeks to emulate natural phenomena, processes, and architectural principles of natural structural and functional materials using advanced tools, sophisticated approaches, and computing technologies [1,2,3]. Extreme biomimetics is a particular direction within biomimetics that was proposed in 2010 [4] and initially aims to harness the mechanisms behind biomineralization processes in organisms that survive in biologically extreme environments [5]. The remarkable ability of living organisms not only to survive but also to flourish in harsh or extreme surroundings has fascinated and motivated scientists across diverse fields of research, including bioinspired materials science. Investigating the underlying mechanisms behind biomineralization strategies in pro- and eukaryotic extremophiles within the frameworks of extreme and forced biomineralization [5] as well as in “macrobiomineral”-producing animals [6] and applying this understanding in the laboratory to create novel hybrid materials has emerged as a central driving factor behind the recent advances in classical biomimetics. By utilizing temperature-, pressure-, and chemical-resistant biopolymers found in these environments, researchers can prepare new inorganic–organic hybrid materials in vitro. As recently proposed by us, the philosophy of extreme biomimetics is partially based on four general approaches:• “Finding corresponding natural sources and examples for inspiration;• Understanding the principles and mechanisms of biological phenomena occurring under natural extremes;• Application of already proven technologies related to the use of biological materials;• Making scientifically based but daring experimental decisions including the development of a new generation of composite materials” [4].

Special attention is given to renewable biopolymers, to avoid depletion of natural resources. One such recognized biological material is spongin, the main protein-based and halogenated skeletal compound found in diverse keratosan marine demosponges [7]. It is a naturally 3D prefabricated microporous biocomposite with a complex fibrous structure that is responsible for the existence of diverse bath sponge skeletons up to 70 cm in diameter [8,9]. Spongin is incredibly robust, with exceptional mechanical strength and resistance to diverse acids and enzymes [7,10,11]. It can withstand high temperatures of up to 360 °C [12], making it an excellent material for use in extreme biomimetic applications under hydrothermal synthesis conditions. The exceptional stability of spongin in the form of 3D microporous scaffolds with high concentrations of metal ions (such as Cu and Fe) opens a key pathway to the development of novel functional composite constructs [13].

Certain species of marine sponges, belonging to the subclass Keratosa in the class Demospongiae, take part in biomineralization involving the formation of iron oxides (such as lepidocrocite, γ-FeO(OH)) on spongin due to the biocorrosion of iron constructs located in seawater near the habitats of these organisms (Figure 1; for an overview see [10,13]). This intriguing relationship between iron ions in seawater and living bath sponges, which are able to transform them into mineral phases, has been an inspiration for our previous research. Consequently, a corresponding biomimetic approach used to synthesize an “Iron–Spongin” composite from spongin and lepidocrocite, exhibiting magnetic properties, has recently been reported [13]. This composite material has demonstrated outstanding performance in dopamine (DA) sensing. Moreover, the iron–spongin material is straightforward to produce and is inexpensive, making it an ideal choice for creating novel biosensors. Spongin-based composites have also achieved successful outcomes in sensing glucose [14], *Staphylococcus aureus* [15], and gallic acid [16].

Iodine is originally present in spongin in concentrations of up to 3%, and the term “iodospongin” has been proposed [9]. However, its possible role in iron mineralization on and within spongin fibers in sponges has not previously been suggested.

Intriguingly, ferrous surfaces undergo very rapid corrosion in reaction with iodine particles, especially in the presence of moisture. Iodine is a strong oxidant, which means that it has a high affinity for electrons and can readily accept them from other substances. When iodine molecules come into contact with iron or steel surfaces, they can oxidize iron atoms, causing the formation of iron iodides. This chemical reaction weakens the structure of iron, making it more susceptible to corrosion (for details see [17,18,19,20,21]).

In this study, an extreme biomimetics approach leading to a novel method for the rapid in vitro formation of goethite (α-FeO(OH)) on a spongin scaffold under laboratory conditions was employed for the first time. The experiment involved reacting crystalline iodine and powdered iron in the presence of selected spongin scaffolds, resulting in the formation of a new composite material named “FeISpongin”. The proposed approach successfully preserves the macroscopic 3D structure of the spongin while producing a multifunctional material that has the potential for large-scale applications.

## 2. Materials and Methods

### 2.1. Materials

Purified spongin scaffolds from the marine demosponge *Hippospongia communis* (Lamarck, 1814) were acquired from INTIB GmbH (Freiberg, Germany). Iron powder (99.99%, with particle sizes in the range of 25–100 µm) and crystalline iodine (99.8%) were purchased from Chempur (Piekary Śląskie, Poland). Goethite standard, dopamine (DA), paraffin oil, and sodium phosphate (Na_2_HPO_4_ and NaH_2_PO_4_) were obtained from Sigma-Aldrich (Burlington, MA, USA). Phosphate buffer solution (0.1 M, pH 6.5) was prepared from a mixture of stock solutions (NaH_2_PO_4_ and Na_2_HPO_4_) and used as an electrolyte solution for amperometric measurements. Graphite powder was obtained from Merck (Darmstadt, Germany).

### 2.2. Sample Preparation

#### Preparation of Materials

The marine sponge (*H. communis*) mineral-free spongin-based skeleton was divided into several parts. Two of these, weighing 1.1 g (for FeISpongin) and 1.2 g (for the control sample) were placed in a 250 mL bottle of distilled water for 10 min. Excess water was then squeezed out of both so that they were slightly damp. The experimental (FeISpongin) and control scaffolds were placed in a glass bottle and shaken vigorously with 3 g iron powder for 10 min. After this time, 3 g of powdered iodine crystals was added to the experimental sample and shaken for 5 min at room temperature. The experimental and control samples were left for 72 h and were then dried and ultrasonically treated for 2 h at room temperature to remove excess powdered iron and iodine crystals (Figure 2).

### 2.3. Characterization Techniques

#### 2.3.1. Digital Microscopy

The materials obtained (FeISpongin and the control sample) were observed and analyzed using an advanced imaging system VHX-6000 digital optical microscope (Keyence, Japan) with a VH-Z20R zoom lens (magnification up to 200×), as well as a VHX-7000 digital optical microscope (Keyence, Japan) with VHX-E20 (magnification up to 100×) and VHX-E100 (magnification up to 500×) zoom lenses.

#### 2.3.2. Scanning Electron Microscopy (SEM) with Energy Dispersive X-ray Analysis (EDX)

To determine the elemental composition and surface morphology of the FeISpongin samples, SEM-EDX measurements were carried out using a low vacuum scanning electron microscope of the type JEOL JSM-6610LV with LaB6 cathode, which was also equipped with an energy dispersive X-ray spectrometer (10 mm^2^ Silicon Drift Detector (SDD) X-Flash 6|10, Bruker Co., Berlin, Germany).

#### 2.3.3. Fourier Transform Infrared Spectroscopy

FTIR spectra of the examined materials were obtained using a Nicolet iS50 spectrometer (Thermo Fisher Scientific Co., Hillsboro, OR, USA). Each measurement was recorded using a built-in attenuated total reflectance (ATR) accessory. The analysis was carried out in a wavelength range of 4000–400 cm^−1^.

#### 2.3.4. Raman Spectroscopy

Spectra were acquired with a confocal Raman microscope (Alpha 300S, WITec GmbH, Ulm, Germany) coupled to a Raman spectrometer (UHTS 300S, WITec GmbH) using laser excitation at 780 nm with TEM00 quality (TA Pro, Toptica Photonics AG, Gräfelfing, Germany). A 50× magnification objective with NA = 0.75 was used to focalize the excitation and collect the Raman signal in a reflection configuration. Raman spectra were punctually recorded using a laser power of about 3 mW, an integration time of 5 s, and an average of 30 spectra to improve the signal-to-noise ratio. Further smoothing was obtained in Matlab using the function ‘mssgolay’.

#### 2.3.5. X-ray Diffraction

X-ray studies of the control and FeISpongin samples were performed using a powder diffractometer (SmartLab Rigaku, Tokyo, Japan) with a CuK alpha lamp, in a 2-theta range of 3–80 (scan step 0.01, scan speed 4°/min).

#### 2.3.6. Thermogravimetric Analysis

Thermogravimetric analysis (TG/DTG) of the materials obtained was carried out on a TGA/DSC1 Star System analyzer (Mettler Toledo, Columbus, OH, USA). Measurements were conducted at a heating rate of 10 °C/min under nitrogen flow conditions (60 mL/min) in a temperature range of 30–700 °C.

#### 2.3.7. Magnetic Properties

The magnetic properties of the control and FeISpongin materials were tested using a neodymium magnet with a pull force of 192 N, purchased from Mistral, Jaworzno, Poland.

#### 2.3.8. Confocal Micro X-ray Fluorescence Spectroscopy (CMXRF)

CMXRF measurements were performed with a modified M4 Tornado commercial MXRF spectrometer (Bruker Nano GmbH, Germany) equipped with a 30 W Rh-microfocus X-ray tube (50 kV, 600 µA), a polycapillary full lens in the excitation channel for X-ray focusing, and a 30 mm^2^ silicon drift detector (SDD). The modification included the installation of a polycapillary half lens in the detection channel in front of a 60 mm^2^ SDD. The confocal arrangement of two lenses resulted in a defined probing volume, providing three-dimensional resolved element analysis by lateral movement of the sample with an xyz motorized sample stage. Calibration of the optics alignment was achieved by precise movement of the second lens using piezo actuators and tracking of the signal intensity of a 2 µm thick Cu foil.

CMXRF measurements were made within a total sample volume of 500 × 500 × 500 µm^3^ and a global step size of 5 µm. A spot measure time of 10 ms was utilized with five measure cycles, resulting in a measure time of 50 ms for each point and an overall measurement time of approx. 63 h. Additionally, with regard to the presence of light elements in the spongin samples, a vacuum of 20 mbar was applied for all measurements.

For initial data evaluation of the 101 generated xy area mappings at varying z positions, the corresponding spectrometer software was used, providing impulse count values for the element signals S-Kα (2.307 keV), I-Lβ (4.239 keV), Fe-Kα (6.397 keV), and Br-Kα (11.902 keV). Due to the physical properties of the lenses, quite different probing volume sizes had to be considered for the different fluorescence energies of the element lines. For the setup used, the probing volume sizes were calculated as a function of the energy by calibrating the spectrometers’ characteristic parameters [22]. Consequently, the following approximate probing volume z-sizes can be expected: S-Kα (69.0 µm), I-Lβ (51.8 µm), Fe-Kα (42.0 µm), and Br-Kα (31.4 µm).

The exported measurement datasets (containing information about the location coordinates x and y and the signal count values) were then further processed using in-house software as used in [13,23], providing tools such as normalization of the xy mappings to a global signal maximum, generation of RGB color-coded images, signal noise correction and stacking of the two-dimensional distribution datasets. The final volume rendering was carried out with the Python application Mayavi [24], leading to three-dimensional distribution images. For the three-dimensional reconstruction of the element distributions—S (yellow), I (magenta), Fe (red), and Br (green)—a volume module was used in combination with light and shade calculations for better visibility of the three-dimensional structure.

Due to the small size of the sponge structure (~30 µm) relative to the probing volume sizes (≥31.4 µm), the properties of natural samples (varying density, elemental composition, absorption due to 3D structure) and different physical behaviors of the observed elements (fluorescence yield, sensitivity, concentration), weak signal values were removed from the volume rendering by setting the alpha values to zero. Consequently, data points were excluded within a range of <4% up to <15% of the global maximum count value, to obtain a less cluttered representation of the 3D elemental distributions. Therefore, the volume reconstructions depict only a qualitative approximation of the 3D elemental distribution. Further data processing is needed for correction of the influence of probing volume size and absorption effects. Since these samples have a quite complex three-dimensional structure and composition, the feasibility of such complex reconstruction tasks (qualitatively and quantitatively) needs to be addressed in future work.

### 2.4. Dopamine Detection

Modified carbon paste electrodes (CPEs) were fabricated by grinding graphite, paraffin oil as a binder, and the modifier in a mortar at a ratio of 65:15:20 (*w*/*w*/*w*) for a time of 40 min. The components were homogenized to form a paste, which was then pressed into a holder with an inner diameter of 4 mm. The prepared electrodes were denoted as Natural-Fe-Spongin/CPE (natural deposition of Fe-oxide on spongin–lepidocrocite) and FeISpongin/CPE (extreme biomimetic deposition of Fe-oxide on spongin–goethite). Amperometric measurements were carried out using a PalmSens 4 electrochemical analyzer and a three-electrode setup with modified CPE as the working electrode, a Ag/AgCl (3 M KCl) reference electrode, and a platinum wire as the counter electrode. The amperometric response of the different modified CPEs for successive addition of DA in 0.1 M phosphate buffer (pH 6.5) was recorded at a potential of 0.25 V.

## 3. Results

### 3.1. Digital Microscopy

Images obtained by digital microscopy show a control sample and the developed FeISpongin composite before and after ultrasound treatment. On the control sample (Figure 3A), iron deposition occurred in microclusters. Small concentrations of reddish color are also visible, which may indicate the formation of iron oxides. No change was observed after 2 h of ultrasound treatment of the control sample (Figure 3B). On the FeISpongin sample (Figure 3C and Figure 4A,B), the spongin fibers are densely coated with a reddish iron oxide layer and black-colored residues of iron and iodine. After ultrasonic treatment of this sample (Figure 3D and Figure 4C), a uniform reddish coloration of the spongin fibers is observed, with no visible residues that have not adhered to the spongin scaffold.

### 3.2. Scanning Electron Microscopy (SEM) with Energy Dispersive X-ray Analysis (EDX)

SEM images were obtained for the FeISpongin sample after ultrasound treatment (Figure 5). A microfiber network forms unique porous structures (Figure 5A) that are typical of spongin scaffolds of *H. communis* origin [13]. Images of the FeISpongin composite show the presence of inorganic clusters (Figure 5B,C). In a near view, crust-like structures can be clearly identified (Figure 5D,E). The high quality of the inorganic layer in the FeISpongin sample demonstrates that the reaction of iron and iodine in the presence of the spongin scaffold results in the deposition of iron oxide crystals. Importantly, the complex porous structure with numerous iron oxide clusters is conserved even after ultrasonic treatment. The weight concentration of iron on the surface of the fiber varies from 32% to 66% (Figure 6, Spots 1, 2, 3, 4, 5, 7, 8) and decreases in the inner layers (Figure 6, Spot 6) (see also Appendix A). Moreover, the composite layer has high values of iodine (from 8% to 19%) as well as oxygen and carbon (Figure 5). In the control sample, a trace iron content (0.8 at%) was detected. This confirmed the formation of crystals consisting mainly of iron during the reaction between iron and iodine in the presence of a spongin scaffold.

### 3.3. Fourier Transform Infrared Spectroscopy (FTIR)

FTIR spectra of the materials were obtained to investigate the presence of characteristic functional groups through which the iron oxide layer formed on the spongin scaffold can be identified. Detailed studies were carried out for the control and FeISpongin samples after ultrasonic treatment (Figure 7). FTIR spectra of the goethite standard were also taken for reference. Details of the bands initially appearing in the spectra, with their wave numbers and band assignments, are presented in Table 1.

The bands in the FTIR spectra of both the control and FeISpongin samples correspond to bands typical for the spongin scaffold, at 3300, 1633, 1536, and 1244 cm^−1^ [25,26,27]. The bands that occur only in the iron/iodine-treated sample are at 3140, 1021, 892, 794, and 635 cm^−1^. The bands around 892, 794, and 635 cm^−1^ are attributed to characteristic vibrations in goethite (α-FeOOH) (Figure 6) [27,28]. This is confirmed by the goethite standard infrared spectrum, where the same bands are observed. In the region of 3142 cm^−1^, there is a typical O–H stretching vibration band for oxyhydroxides [29]. Enhancement of the bands near 565–700 cm^- 1^ attributed to Fe–O stretching vibrations in goethite can also be observed [30]. Further significant bands around 794 and 892 cm^−1^ result from in-plane deflection of surface OH in Fe–OH–Fe [31]. The band at 1021 cm^−1^ in the FTIR spectrum of FeISpongin may indicate the additional presence of traces of lepidocrocite (γ-FeOOH), which is a characteristic iron oxide for the spongin scaffold [10,32].

### 3.4. Raman Spectroscopy

Raman spectroscopy was used to determine the qualitative properties of the crystalline product in the FeISpongin composite material (Figure 8). All bands present on this spectrum correspond to those assigned to goethite in the literature [33,34,35]. The strongest band is observed at 387 cm^−1^; these vibrations can be characterized as a mixture of Fe–O–Fe bond angle bending and Fe–O symmetric stretching in goethite [36]. In addition, a transition to hematite was observed at a laser power higher than the one used for the measurement, consistent with the well-known dehydration effect of goethite upon heating [33].

### 3.5. X-ray Diffraction

The X-ray diffraction pattern of pure spongin lacks strong peaks, which is in accordance with previously reported findings [25]. The X-ray diffractogram of the control sample closely resembles that of the spongin sample, with no peaks corresponding to iron-containing biominerals. However, when the spongin sample was treated with a combination of iron and iodine, distinct reflections characteristic of goethite (α-FeOOH) were observed [37,38,39,40], indicating that this mineral is produced during the fabrication process of the FeISpongin composite. This is supported by the detection of characteristic peaks for this biomineral at approximately 21°, 33°, 34°, 36°, 41°, 53°, 59°, and 61°, corresponding to the crystal planes (110), (130), (021), (111), (140), (410), (151), and (002). To aid in the comparison, a diffractogram of the goethite standard is also presented (Figure 9).

### 3.6. Thermogravimetric Analysis

Thermal degradation of the goethite standard and the obtained samples after ultrasound treatment was also studied. Two mass losses were observed during the thermal degradation of the samples (Figure 10). The first, in the range of 50–150 °C, is related to the evaporation of physically adsorbed water and hydrogen-bonded water [8,41,42]. The second mass loss in the spongin-based samples, occurring in the temperature range of 250–470 °C, may be related to the thermal degradation of peptide bonds [43] and the breakdown of disulfide bonds [8,44] and hydrogen bonds [8]. It should be noted that spongin contains up to 5% sulfur of organic origin [9]. In the goethite standard, a mass decrease of around 250 °C is associated with the transformation to hematite [42,45,46].

The FeISpongin material exhibited higher thermal stability than the control sample, this being attributable to the bonds formed between spongin and iron and the electrostatic interactions between the hydroxyl groups of spongin and iron oxide [26]. This confirms the effectiveness of the method applied to obtain a new goethite–spongin composite material using crystalline iodine.

### 3.7. Magnetic Properties

As Figure 11 shows, the FeISpongin composite material is attracted by a neodymium magnet with a pull force of 192 N. Goethite at room temperature is antiferromagnetically ordered with a Néel temperature of about 120 °C [36,47]. Although it is considered to be antiferromagnetic (AFM), a number of authors report that goethite has a magnetic component, generally described as weak ferromagnetism (WFM) [48,49,50,51,52,53]. There are several hypotheses in the literature as to the origin of this magnetic component, possibilities include breaks in Fe–O chains [52], excessive OH^-^ resulting in the creation of vacancies [51], and finite size effects [54].

### 3.8. Confocal Micro X-ray Fluorescence Spectroscopy (CMXRF)

The iron-treated spongin sample, FeISpongin (produced by reaction with iron powder and iodine, followed by cleaning in distilled water with ultrasonication) exhibited a less even structure than the pure spongin control sample [13]. Higher maximum count values were detected for Fe (Fe-Kα: ≤1493.0 counts vs. ≤18.0) and I (I-Lβ: ≤82.0 counts vs. ≤19.0 counts). The maximum values in particular can be attributed to residue particles within the sponge structure (see the dark particles in the marked mapping area in the video image (Appendix A) and Fe-Kα and I-Lβ distributions in Figure 12), which result from the reaction of iron and iodine at the surface of the spongin and are probably iron powder residues which have not been completely removed after the washing step. Counts of the elements S and Br are comparable to those obtained from the pure spongin sample (S-Kα: ≤19.0 counts vs. ≤ 18.0; Br-Kα: ≤20.0 counts vs. ≤19.0); they are mainly detected within the sponge structure. However, calcium was not detected in the FeISpongin sample despite its distinct count values in the pure spongin sample (Ca-Kα: ≤19.0 counts).

### 3.9. Dopamine Detection

The detection of DA, a neurotransmitter that significantly influences the cognitive and behavioral activities of living organisms, is a task of crucial importance. Monitoring irregularities in DA levels in the human body might aid the early detection of neurological illnesses such as Parkinson’s, Alzheimer’s, schizophrenia, etc. [55,56]. Several analytical techniques have been applied to detect DA; however, each has some drawbacks. In the presence of other biological substances, electrochemical approaches are the most effective way to determine DA [57,58,59]. Developing a simple, cost-effective composite as an electrode material for the selective detection of DA at low concentrations in the absence of interference from other biological substances remains a challenge.

In this study, for the first time, a novel, low-cost, sensitive, and selective electrochemical sensor for the detection of DA based on CPEs modified with spongin–Fe-oxide was developed. The α- and γ-crystalline phases of iron oxyhydroxide (FeOOH) were formed on the 3D spongin scaffold through natural and extreme biomimetic processes, respectively. To assess the sensitivity of FeOOH electrocatalysts, amperometry tests were performed for both prepared systems, Natural-Fe-Spongin/CPE and FeISpongin/CPE, as shown in Figure 13A and Figure 13B, respectively. The oxidation potential (0.25 V) was applied while different concentrations of DA were injected at regular intervals into 0.1 M phosphate buffer (pH 6.5). The calibration curves (Figure 13A,B inset) recorded the rise in the current with each addition of DA. The linear regression equation of DA oxidation for each system was obtained between 5 μM and 1.3 mM; the equations were I(µA) = 28.104 CDA (mM) + 0.7336 (R² = 0.998) for Natural-Fe-Spongin/CPE, and I(µA) = 26.658 CDA (mM) + 1.6267 (R² = 0.9948) for FeISpongin/CPE. The sensitivities of Natural-Fe-Spongin/CPE and FeISpongin/CPE were found to be 0.22 and 0.21 μA μM^−1^ cm^−2^, respectively. The enhanced electrocatalytic activity of both α- and γ-crystalline phases of FeOOH in DA sensing is attributed to the high numbers of active sites of Fe-oxide adsorbed on a 3D spongin scaffold and their facile charge transfer characteristics.

The selectivity of the FeISpongin/CPE sensor was investigated in the presence of potential coexisting species (sucrose, glucose, NaCl, and UA). The results demonstrated that the sensor reduced the effect of potential interfering species and achieved outstanding DA detection selectivity. Due to the effects of aberrant DA concentrations on the regulation of blood pressure, lipolysis, Huntington’s disease, and Parkinson’s disease, the detection of DA in human urine is of interest in medical diagnostics. FeISpongin/CPE was used to detect DA in human urine samples in order to evaluate the practical usability of the developed DA sensor. The recovered sample ranged from 96% to 116%, indicating the precision of the developed sensor, which implies its suitability for on-site usage.

## 4. Discussion

When considering the effect of the presence of iodine on the rapid formation of iron minerals on spongin fibers, it is worth noting that the natural skeleton of the marine sponge contains a significant amount of organic iodine. In 1819 [60], Andrew Fyfe made the first report concerning substantial quantities of iodine found in the *Spongia usta* demosponge, commonly known as the “Coventry Remedy”, which had been widely used in ancient China. Hundeshagen [61] later (in 1895) discussed several sponges rich in iodine known as “Iodospongin”. Harnack [62] hypothesized an organic source for the sponge’s iodine content. The concentration of iodine in the sponge was estimated to be between 1.1% and 1.2%. It was demonstrated that superheated steam completely destroys the organic part of the spongin fibers, releasing iodine. In 1898 [61], “Iodospongin” was isolated and characterized as an albumin-like product containing over 8.5% iodine and 9.4% nitrogen. Bath sponges were extensively researched as a source of iodine until 1914.

Our previous studies [13] have indicated a correlation between iron mineral formation on spongin fibers in seawater and the amino acid sequences present—specifically cysteine, histidine, lysine, or tyrosine. These amino acid functional groups containing electron donor atoms, such as sulfur, nitrogen, and oxygen, enable the creation of iron ion complexes. Cysteine’s ability to oxidatively dehydrate, form disulfide bonds, and facilitate the transformation of iron phases likely explains its involvement in the formation of iron-based crystalline mineral phases. More stable iron mineral phases, such as lepidocrocite, are developed through the reduction of iron sites and dissolution/precipitation processes. Lepidocrocite, accompanied by small amounts of goethite, is commonly found in the natural spongin fibers of the *Spongia officinalis* demosponge [63]. It should be noted that goethite is a naturally occurring mineral and a significant component of rust on metal structures in standard environmental conditions [64].

In the field of interactions between metal and halogen, the tendency of iodine to form compounds with iron has received considerable attention. Iodine is a typical halogen and demonstrates a marked ability to participate in redox reactions, primarily functioning as an oxidizing agent. This behavior of iodine plays a significant role in the occurrence of iodine-assisted corrosion [65,66]. When iodine interacts with iron, it forms an iron(II) iodide. When exposed to atmospheric oxygen, it is hypothesized that this iron–iodine compound could undergo a redox change [65]. The initial divalent state of the iron could change to a trivalent state, potentially forming structures reminiscent of ferrihydrite. However, while ferrihydrite is recognized as an aqueous iron oxide and a potential precursor to more stable minerals such as goethite [67,68], it is important to note that its formation in this context remains speculative and warrants detailed investigation.

At the same time, we expect that the wet spongin scaffold may play a key role in the emerging chemical dynamics. This scaffold is not simply a static foundation but has a complex organic structure and a diverse chemical composition. These features have led us to theorize its potential dual role: as a catalyst and as a navigational force directing these reactions. In particular, the structure provides an environment in which iodine and iron can interact closely, potentially directing the course of the reactions. There are many known iron oxides and hydroxides, but goethite stands out for its well-known thermodynamic stability [69]. With this in mind, it has been hypothesized that a progression from a ferrihydrite-like unit to the final crystalline structure of goethite may occur under such conditions (Figure 14). However, the exact nature and occurrence of this transition remains the subject of further research and validation.

Goethite is a prevalent mineral found in soils, ores, sediments, and other environments [70]. Both natural and laboratory-synthesized goethite contain nanometer-sized particles, with lengths ranging from several microns to nanometers. This results in a high specific surface area that can range from 10 to 132 m^2^g^−1^ [71,72] depending on the transformation environment and synthesis conditions. This high surface area makes goethite a promising candidate for use as an adsorbent, catalyst, or sensor.

There are a number of industrial applications for goethite, which require the use of a variety of methods for its preparation. The most frequently used technique is hydrothermal synthesis [73]; however, other techniques such as sol–gel [74], forced hydrolysis [75], and precipitation methods [76] are also adopted. A particularly interesting approach is the use of an organic matrix for the nucleation and growth of α-FeOOH. A composite of goethite and plant-fibre loofah sponge was prepared via a mechanistic solid-phase technique [77]. The composite achieves a notably high adsorption capacity (six times greater than that of pure goethite) and possesses visible-light photocatalytic ability. As such, this material is highly promising in terms of its potential to remove organic pollutants in a very efficient and environmentally friendly manner [77].

Marine sponges offer biodegradability, sustainable resources, potential plastic replacement, ecosystem support, and lasting environmental impact. They are renewable sources of spongin, due to marine technology that enables cultivation on an industrial scale. The natural properties of goethite combined with those of spongin, including exceptional mechanical strength, thermal stability at temperatures up to 360 °C, and high resistance in the presence of various proteolytic enzymes and corrosive acids, create remarkable composites that will open up many opportunities in areas such as biomedicine, agriculture, and the restoration of our environment.

One potential application may be as a biosensor for neurotransmitters. Electrochemical methods are being explored as a potential way to detect neurotransmitters like DA, with advantages including high sensitivity, quick response time, low cost of materials, and practicality [78,79]. Enzymatic biosensors have limitations due to enzyme instability, and for this reason, non-enzymatic sensors based on metal oxides, such as iron, have been developed. These sensors are portable, simple, and fast, and can separate analytes through magnetic properties [80,81,82]. Magnetic iron oxide and composites based on the natural polymer spongin show promise as non-enzymatic sensors for detecting neurotransmitters. Our results showed that the FeISpongin composite used as a sensor exhibited exceptional selectivity for DA detection, as well as enhanced electrocatalytic activity.

The magnetic properties of the FeISpongin composite material warrant further investigation. Future studies can perform a thorough analysis of its magnetic behavior using techniques such as magnetic susceptibility measurements [83], hysteresis loop analysis [84], and Mössbauer spectroscopy [85]. This investigation will facilitate the identification of the origin and implications of the magnetic behavior that has been observed. Additionally, it will enable the study of the relevance of the observed magnetic behavior to the overall properties and potential applications of the composite material.

The uniqueness of the goethite–spongin composite we designed lies in its three-dimensional architecture while maintaining microporosity. Using the already known functional features of this mineral phase as a catalyst [86,87]; sorbent in wastewater treatment [88]—including removal of dyes [89] and heavy metal ions [90,91]; antibacterial agent [92]; stabilator of diverse enzyme–mineral [93], as well as amino acids–mineral complexes [94]; and functional material in biomedicine [95] and modern dentistry [96], the development of appropriate systems should be undertaken in the near future.

## 5. Conclusions

Extreme biomimetics undoubtedly represents an intriguing frontier in scientific research, where the aim is to replicate natural processes, but with an emphasis on extreme conditions. In this field, scientists are interested in not only mimicking natural solutions but also using corresponding biopolymers in artificial environments that are considered challenging, unusual, or extreme. Spongin, a robust marine biomaterial, already plays a key role in the creation of inorganic–organic hybrid materials for many applications. In our study, iodine, known for its strong oxidizing properties, was used as a tool to induce rapid corrosion of iron microparticles, leading to the development of the FeISpongin composite. This composite, which combines goethite with spongin, displays potential in a variety of fields, including biomedicine and environmental restoration. In particular, it shows promise as a non-enzymatic biosensor of neurotransmitters such as DA.

## Figures and Tables

**Figure 1 biomimetics-08-00533-f001:**
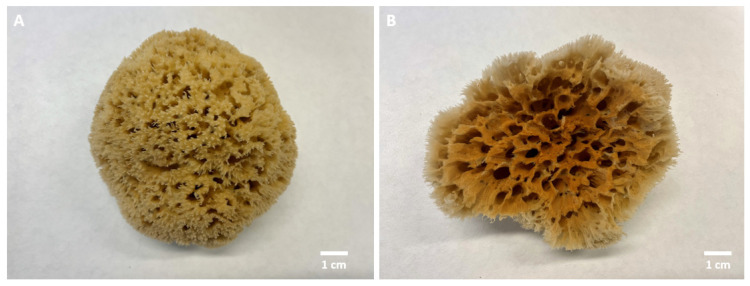
The natural skeleton of the marine demosponge *Hippospongia communis*, when cultivated in the absence of iron ions, displays a yellowish tint (**A**). When iron ions are present it undergoes a noticeable transformation, taking on a prominent rust-colored appearance due to the presence of lepidocrocite (**B**). For details see [13].

**Figure 2 biomimetics-08-00533-f002:**
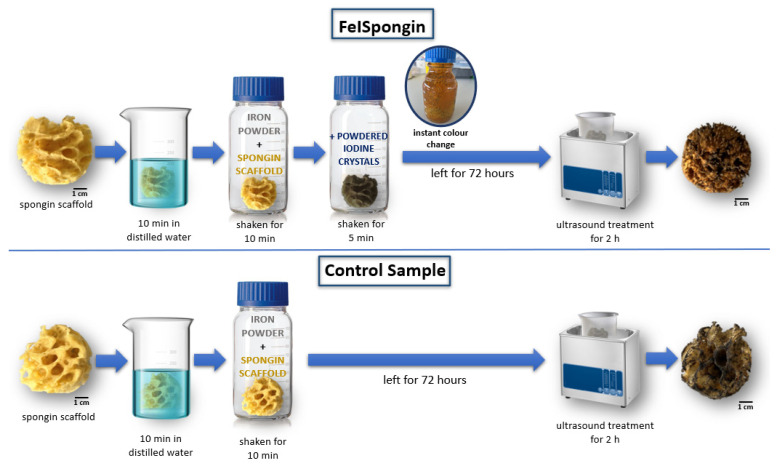
Schematic illustration of the preparation of materials for the study.

**Figure 3 biomimetics-08-00533-f003:**
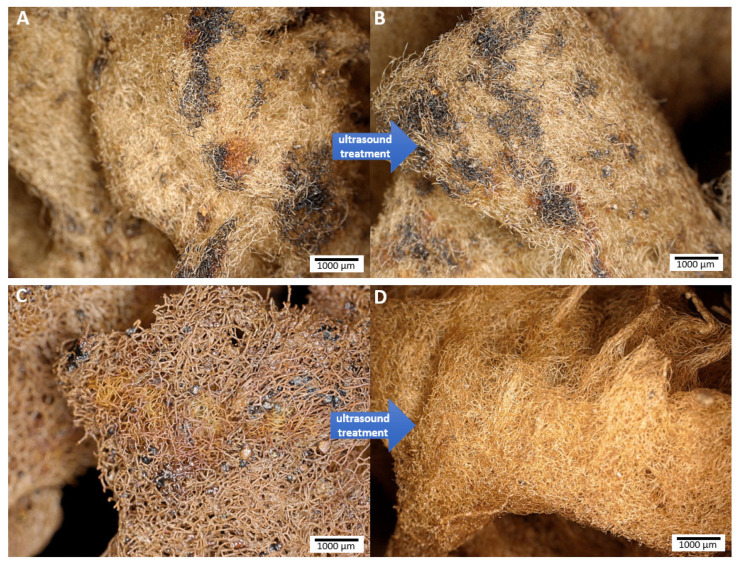
Digital microscopy imagery: (**A**,**B**) control sample; (**C**,**D**) FeISpongin composite before and after ultrasound treatment.

**Figure 4 biomimetics-08-00533-f004:**
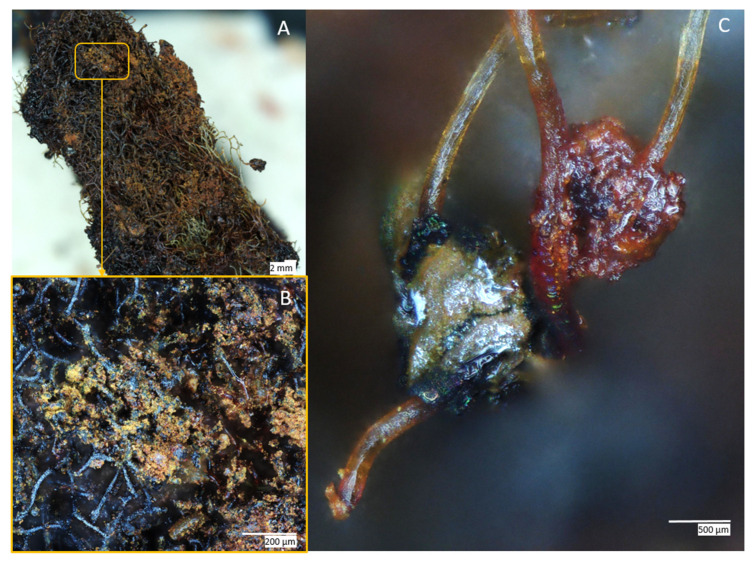
Digital microscopy of the FeISpongin composite formed on the *H. communis* spongin scaffold: (**A**) dried scaffold sample after composite formation; (**B**) yellowish-brown to dark brown and black incorporations are visible on the spongin scaffold; (**C**) composite particles stay fixed on the spongin fibers after 2 h of ultrasound treatment.

**Figure 5 biomimetics-08-00533-f005:**
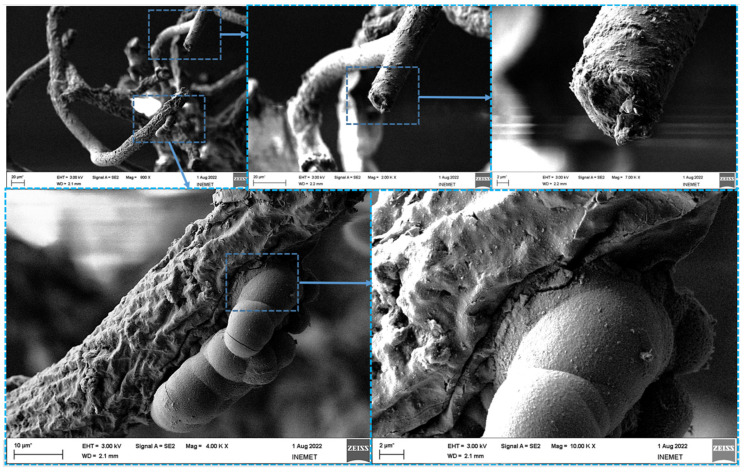
SEM imagery of FeISpongin composite: (**A**) spongin scaffold, (**B**,**C**) crust-like structure covering spongin fiber, (**D**,**E**) clearly visible iron mineral conglomerates built on spongin fiber.

**Figure 6 biomimetics-08-00533-f006:**
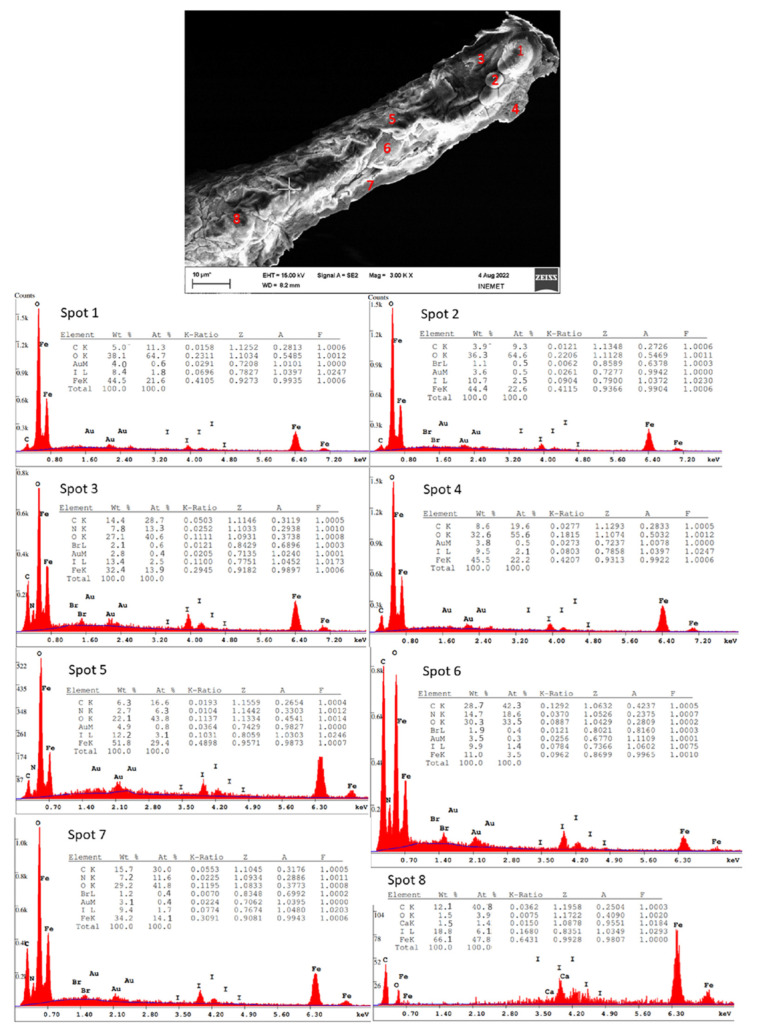
SEM image with EDX quantification of selected spongin fiber after formation of FeISpongin composite. The blue line marks mainly the continuous bremsstrahlung X-ray background, which is substracted for the quantification.

**Figure 7 biomimetics-08-00533-f007:**
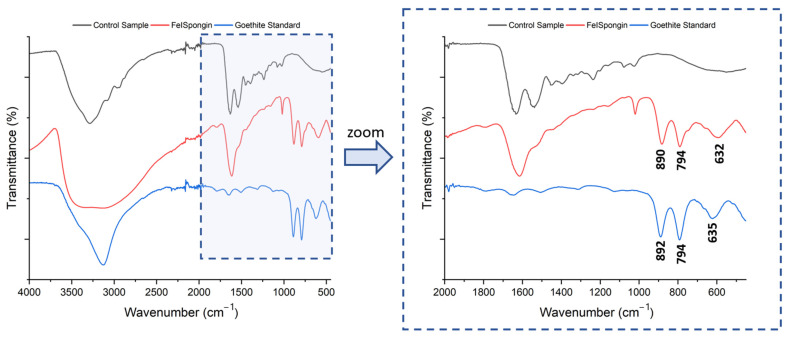
FTIR spectra of the spongin control sample, FeISpongin, and a goethite standard analyzed for comparison.

**Figure 8 biomimetics-08-00533-f008:**
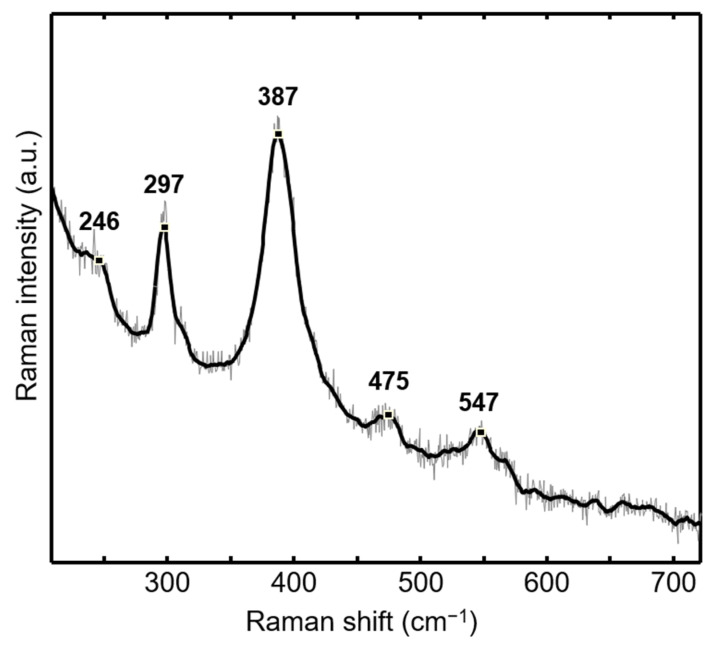
Raman spectra of the FeISpongin composite.

**Figure 9 biomimetics-08-00533-f009:**
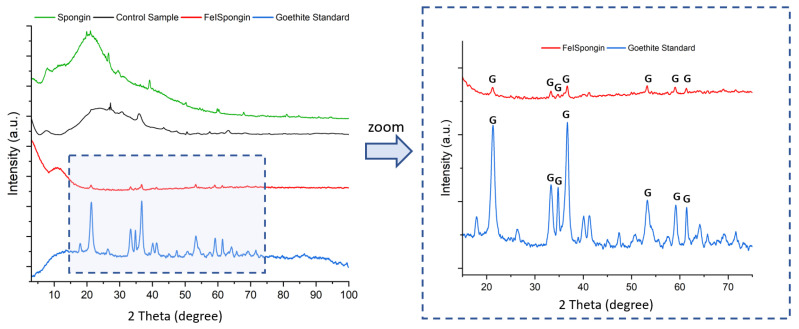
XRD patterns of spongin control sample, FeISpongin composite, and goethite standard.

**Figure 10 biomimetics-08-00533-f010:**
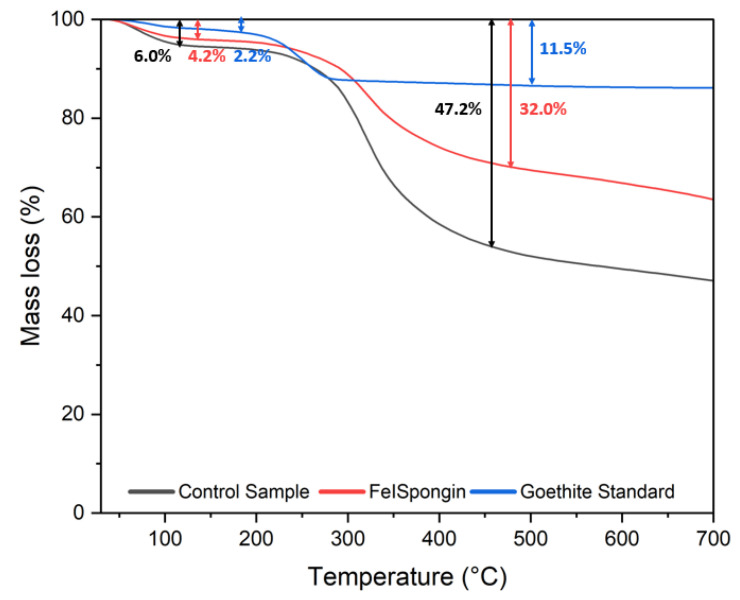
Thermogravimetric (TG) curves of control sample, FeISpongin composite, and goethite standard.

**Figure 11 biomimetics-08-00533-f011:**
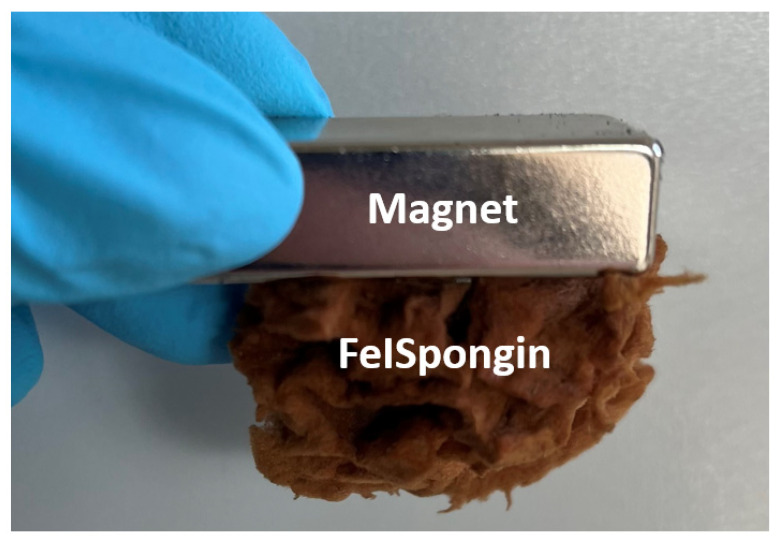
The 3D FeISpongin composite scaffold attracted by a magnet.

**Figure 12 biomimetics-08-00533-f012:**
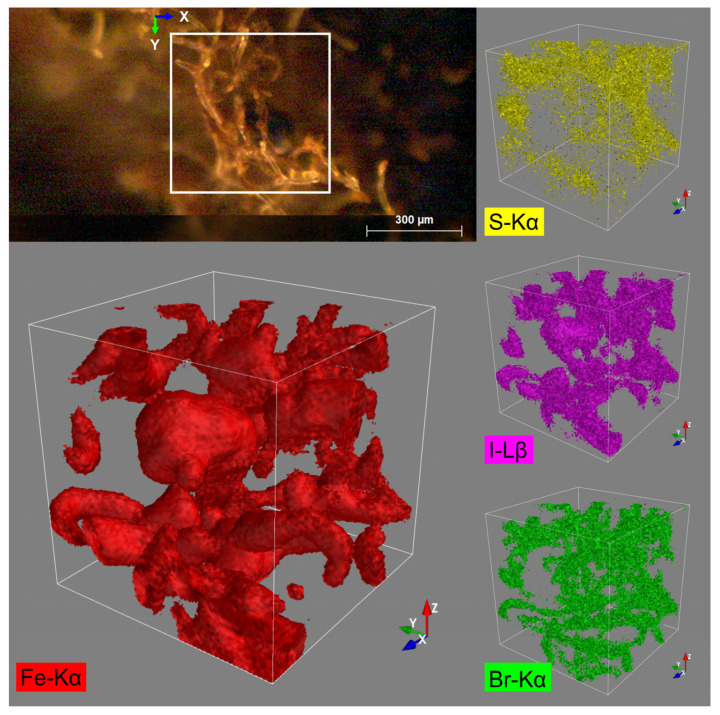
The 3D distribution images of the elements S (Kα), Fe (Kα), Br (Kα), and I (Lβ) within an analyzed volume of 500 × 500 × 500 µm of the FeISpongin sample.

**Figure 13 biomimetics-08-00533-f013:**
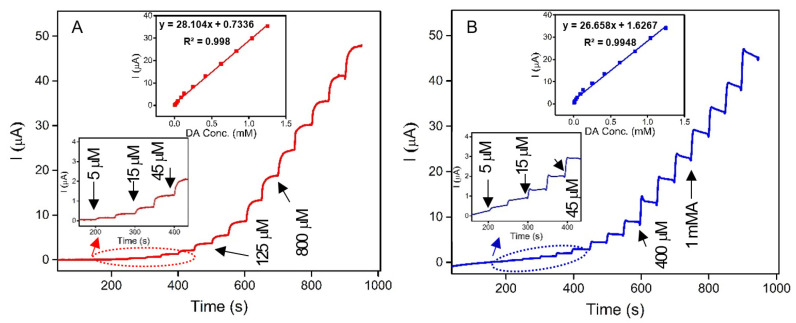
Amperograms recorded in 0.1 M phosphate buffer (pH 6.5) with the successive addition of DA (5 μM to 1.3 mM) at (**A**) Natural-Fe-Spongin/CPE and (**B**) FeISpongin/CPE. Inset: calibration curve for linear response of current vs. DA concentration.

**Figure 14 biomimetics-08-00533-f014:**
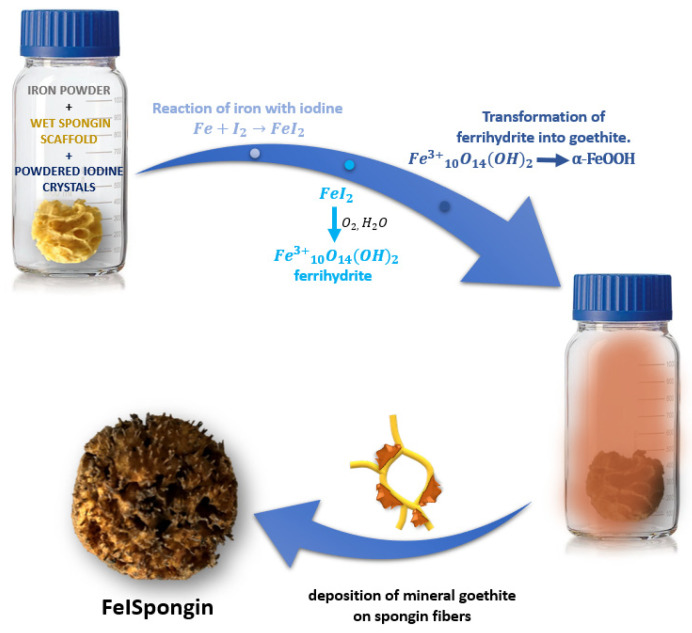
Schematic representation of the possible mechanism of goethite formation on spongin fibers.

**Table 1 biomimetics-08-00533-t001:** Wavenumbers of the bands of the materials under study and their assignment.

Control Sample	FeISpongin	Goethite Standard	Vibrational Assignment
3300	3300	-	–NH stretching
-	3140	3140	–OH stretching
1633	1633	-	C=O stretching
1536	1536	-	–NH deformational
1244	1244	-	C–N stretching
-	1021	-	Fe–OH
-	892	890	–OH bending
-	794	794	–OH bending
	635	632	Fe–O stretching

## Data Availability

The data presented in this study are available on request from the corresponding authors.

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
