# Peer review of "Creation of a 3D Goethite–Spongin Composite Using an Extreme Biomimetics Approach"

_biomimetics, 2023, doi:10.3390/biomimetics8070533_

Round 1

Reviewer 1 Report

Comments and Suggestions for Authors

This is a fantastic work. I have no comments. It should be published as it is. This work represent another great contribution to the field of extreme biomimetics. 

Author Response

Thank you for your positive feedback and recommendation. We greatly appreciate your support in recognizing the value of our work in the field of extreme biomimetics.

Reviewer 2 Report

Comments and Suggestions for Authors

The authors reported Creation of a 3D goethite–spongin composite using an extreme biomimetics approach. The authors created and characterized FeISpongin in detail, showing a satisfactory application in the electrochemical sensing detection of dopamine (DA). However, there are so many problems in the manuscript, and I don’t think the manuscript is suitable for publication before modification. The questions as follows:

1.       A space is required between the number and the unit, such as “2h” in Figure 2 and “5µM, 15µM and 45µM” in Figure 13.

2.       The authors in this manuscript provide a more comprehensive characterization of the FeISpongin composite, yet there is limited description and characterization regarding its performance. For example, authors should provide more related data, discussion and potential application about its magnetic properties or other properties.

3.       In addition to DA detection, the potential applications of FeISpongin in other fields, such as regenerative medicine and material science, warrant further discussion.

Author Response

Query:

A space is required between the number and the unit, such as “2h” in Figure 2 and “5µM, 15µM and 45µM” in Figure 13.

Response: Thank you for your valuable comment: we have made corresponding changes in the revised manuscript.

Query:

The authors in this manuscript provide a more comprehensive characterization of the FeISpongin composite, yet there is limited description and characterization regarding its performance. For example, authors should provide more related data, discussion and potential application about its magnetic properties or other properties.

Response: Thank you for your valuable feedback on our manuscript. We appreciate your input and will address the points you've raised. In this study, our primary focus was on the synthesis and characterization of the FeISpongin composite material. We agree that a more thorough examination of its magnetic properties would be a valuable addition. In our future work, we plan to conduct a detailed investigation of the magnetic properties using techniques such as magnetic susceptibility measurements, hysteresis loop analysis, and Mössbauer spectroscopy. These additional experiments will provide a comprehensive understanding of the material's magnetic behaviour. See also revised manuscript ( section 4. Discussion).

Query:

In addition to DA detection, the potential applications of FeISpongin in other fields, such as regenerative medicine and material science, warrant further discussion.

Response:

Thank you for this important remark. Now, we inserted corresponding fragment into the revised manuscript:

"The uniqueness of the goethite-spongin composite we designed lies in its three-dimensional architecture while maintaining microporosity. Using the already known functional features of this mineral phase as catalyst [97,98], sorbent in wastewater treatment [99] including removal of dyes [100] and heavy metal ions [101,102], antibacterial agent [103], stabilator of diverse enzyme-mineral [104] as well as amino acids-mineral complexes [105], functional material in biomedicine [106] and modern dentistry [107] the development of appropriate systems should be undertaken in the near future."